# Deep Graph Infomax

**Petar Veličković**[*]
Department of Computer Science and Technology
University of Cambridge
`petar.velickovic@cst.cam.ac.uk`

**William Fedus**
Mila – Québec Artificial Intelligence Institute
Google Brain
`liamfedus@google.com`

**William L. Hamilton**
Mila – Québec Artificial Intelligence Institute
McGill University
`wlh@cs.mcgill.ca`

**Pietro Liò**
Department of Computer Science and Technology
University of Cambridge
`pietro.lio@cst.cam.ac.uk`

**Yoshua Bengio**[†]
Mila – Québec Artificial Intelligence Institute
Université de Montréal
`yoshua.bengio@mila.quebec`

**R Devon Hjelm**
Microsoft Research
Mila – Québec Artificial Intelligence Institute
`devon.hjelm@microsoft.com`

## Abstract

We present *Deep Graph Infomax* (DGI), a general approach for learning node representations within graph-structured data in an unsupervised manner. DGI relies on maximizing mutual information between patch representations and corresponding high-level summaries of graphs—both derived using established graph convolutional network architectures. The learnt patch representations summarize subgraphs centered around nodes of interest, and can thus be reused for downstream node-wise learning tasks. In contrast to most prior approaches to unsupervised learning with GCNs, DGI does not rely on random walk objectives, and is readily applicable to both transductive and inductive learning setups. We demonstrate competitive performance on a variety of node classification benchmarks, which at times even exceeds the performance of supervised learning.

## 1 Introduction

Generalizing neural networks to graph-structured inputs is one of the current major challenges of machine learning (Bronstein et al., 2017; Hamilton et al., 2017b; Battaglia et al., 2018). While significant strides have recently been made, notably with *graph convolutional networks* (Kipf & Welling, 2016a; Gilmer et al., 2017; Veličković et al., 2018), most successful methods use *supervised learning*, which is often not possible as most graph data in the wild is unlabeled. In addition, it is often desirable to discover novel or interesting structure from large-scale graphs, and as such, unsupervised graph learning is essential for many important tasks.

Currently, the dominant algorithms for unsupervised representation learning with graph-structured data rely on random walk-based objectives (Grover & Leskovec, 2016; Perozzi et al., 2014; Tang et al., 2015; Hamilton et al., 2017a), sometimes further simplified to reconstruct adjacency information (Kipf & Welling, 2016b; Duran & Niepert, 2017). The underlying intuition is to train an encoder network so that nodes that are "close" in the input graph are also "close" in the representation space.

While powerful—and related to traditional metrics such as the personalized PageRank score (Jeh & Widom, 2003)—random walk methods suffer from known limitations. Most prominently, the random-walk objective is known to over-emphasize proximity information at the expense of structural information (Ribeiro et al., 2017), and performance is highly dependent on hyperparameter choice (Grover & Leskovec, 2016; Perozzi et al., 2014). Moreover, with the introduction of stronger

---

[*]Work performed while the author was at Mila.
[†]CIFAR Fellow

encoder models based on graph convolutions (Gilmer et al., 2017), it is unclear whether random-walk objectives actually provide any useful signal, as these encoders already enforce an inductive bias that neighboring nodes have similar representations.

In this work, we propose an alternative objective for unsupervised graph learning that is based upon *mutual information*, rather than random walks. Recently, scalable estimation of mutual information was made both possible and practical through Mutual Information Neural Estimation (MINE, Belghazi et al., 2018), which relies on training a *statistics network* as a classifier of samples coming from the joint distribution of two random variables and their product of marginals. Following on MINE, Hjelm et al. (2018) introduced Deep InfoMax (DIM) for learning representations of high-dimensional data. DIM trains an encoder model to maximize the mutual information between a high-level "global" representation and "local" parts of the input (such as patches of an image). This encourages the encoder to carry the type of information that is present in all locations (and thus are *globally relevant*), such as would be the case of a class label.

DIM relies heavily on convolutional neural network structure in the context of image data, and to our knowledge, no work has applied mutual information maximization to graph-structured inputs. Here, we adapt ideas from DIM to the graph domain, which can be thought of as having a more general type of structure than the ones captured by convolutional neural networks. In the following sections, we introduce our method called *Deep Graph Infomax* (DGI). We demonstrate that the representation learned by DGI is consistently competitive on both transductive and inductive classification tasks, often outperforming both supervised and unsupervised strong baselines in our experiments.

## 2 RELATED WORK

**Contrastive methods**. An important approach for unsupervised learning of representations is to train an encoder to be *contrastive* between representations that capture statistical dependencies of interest and those that do not. For example, a contrastive approach may employ a *scoring function*, training the encoder to increase the score on "real" input (a.k.a, positive examples) and decrease the score on "fake" input (a.k.a., negative samples). Contrastive methods are central to many popular word-embedding methods (Collobert & Weston, 2008; Mnih & Kavukcuoglu, 2013; Mikolov et al., 2013), but they are found in many unsupervised algorithms for learning representations of graph-structured input as well. There are many ways to score a representation, but in the graph literature the most common techniques use classification (Perozzi et al., 2014; Grover & Leskovec, 2016; Kipf & Welling, 2016b; Hamilton et al., 2017b), though other scoring functions are used (Duran & Niepert, 2017; Bojchevski & Günnemann, 2018). DGI is also contrastive in this respect, as our objective is based on classifying local-global pairs and negative-sampled counterparts.

**Sampling strategies**. A key implementation detail to contrastive methods is how to draw positive and negative samples. The prior work above on unsupervised graph representation learning relies on a local contrastive loss (enforcing proximal nodes to have similar embeddings). Positive samples typically correspond to pairs of nodes that appear together within *short random walks* in the graph—from a language modelling perspective, effectively treating nodes as *words* and random walks as *sentences*. Recent work by Bojchevski & Günnemann (2018) uses node-anchored sampling as an alternative. The negative sampling for these methods is primarily based on sampling of random pairs, with recent work adapting this approach to use a curriculum-based negative sampling scheme (with progressively "closer" negative examples; Ying et al., 2018a) or introducing an adversary to select the negative examples (Bose et al., 2018).

**Predictive coding**. Contrastive predictive coding (CPC, Oord et al., 2018) is another method for learning deep representations based on mutual information maximization. Like the models above, CPC is also contrastive, in this case using an estimate of the conditional density (in the form of noise contrastive estimation, Gutmann & Hyvärinen, 2010) as the scoring function. However, unlike our approach, CPC and the graph methods above are all *predictive*: the contrastive objective effectively trains a predictor between structurally-specified parts of the input (e.g., between neighboring node pairs or between a node and its neighborhood). Our approach differs in that we contrast global / local parts of a graph simultaneously, where the global variable is computed from all local variables.

To the best of our knowledge, the sole prior works that instead focuses on contrasting "global" and "local" representations on graphs do so via (auto-)encoding objectives on the adjacency matrix

(Wang et al., 2016) and incorporation of community-level constraints into node embeddings (Wang et al., 2017). Both methods rely on matrix factorization-style losses and are thus not scalable to larger graphs.

## 3 DGI Methodology

In this section, we will present the Deep Graph Infomax method in a top-down fashion: starting with an abstract overview of our specific unsupervised learning setup, followed by an exposition of the objective function optimized by our method, and concluding by enumerating all the steps of our procedure in a single-graph setting.

### 3.1 Graph-based unsupervised learning

We assume a generic graph-based unsupervised machine learning setup: we are provided with a set of *node features*, $\mathbf{X} = \{\vec{x}_1, \vec{x}_2, \dots, \vec{x}_N\}$, where $N$ is the number of nodes in the graph and $\vec{x}_i \in \mathbb{R}^F$ represents the features of node $i$. We are also provided with relational information between these nodes in the form of an *adjacency matrix*, $\mathbf{A} \in \mathbb{R}^{N \times N}$. While $\mathbf{A}$ may consist of arbitrary real numbers (or even arbitrary edge features), in all our experiments we will assume the graphs to be *unweighted*, i.e. $A_{ij} = 1$ if there exists an edge $i \to j$ in the graph and $A_{ij} = 0$ otherwise.

Our objective is to learn an *encoder*, $\mathcal{E} : \mathbb{R}^{N \times F} \times \mathbb{R}^{N \times N} \to \mathbb{R}^{N \times F'}$, such that $\mathcal{E}(\mathbf{X}, \mathbf{A}) = \mathbf{H} = \{\vec{h}_1, \vec{h}_2, \dots, \vec{h}_N\}$ represents high-level representations $\vec{h}_i \in \mathbb{R}^{F'}$ for each node $i$. These representations may then be retrieved and used for downstream tasks, such as node classification.

Here we will focus on *graph convolutional* encoders—a flexible class of node embedding architectures, which generate node representations by repeated aggregation over local node neighborhoods (Gilmer et al., 2017). A key consequence is that the produced node embeddings, $\vec{h}_i$, *summarize a patch* of the graph centered around node $i$ rather than just the node itself. In what follows, we will often refer to $\vec{h}_i$ as *patch representations* to emphasize this point.

### 3.2 Local-global mutual information maximization

Our approach to learning the encoder relies on *maximizing local mutual information*—that is, we seek to obtain node (i.e., local) representations that capture the global information content of the entire graph, represented by a *summary vector*, $\vec{s}$.

In order to obtain the graph-level summary vectors, $\vec{s}$, we leverage a *readout function*, $\mathcal{R} : \mathbb{R}^{N \times F} \to \mathbb{R}^F$, and use it to summarize the obtained patch representations into a *graph-level representation*; i.e., $\vec{s} = \mathcal{R}(\mathcal{E}(\mathbf{X}, \mathbf{A}))$.

As a proxy for maximizing the local mutual information, we employ a *discriminator*, $\mathcal{D} : \mathbb{R}^F \times \mathbb{R}^F \to \mathbb{R}$, such that $\mathcal{D}(\vec{h}_i, \vec{s})$ represents the probability scores assigned to this patch-summary pair (should be higher for patches contained within the summary).

Negative samples for $\mathcal{D}$ are provided by pairing the summary $\vec{s}$ from $(\mathbf{X}, \mathbf{A})$ with patch representations $\vec{h}_j$ of an alternative graph, $(\widetilde{\mathbf{X}}, \widetilde{\mathbf{A}})$. In a multi-graph setting, such graphs may be obtained as other elements of a training set. However, for a single graph, an explicit (stochastic) *corruption function*, $\mathcal{C} : \mathbb{R}^{N \times F} \times \mathbb{R}^{N \times N} \to \mathbb{R}^{M \times F} \times \mathbb{R}^{M \times M}$ is required to obtain a negative example from the original graph, i.e. $(\widetilde{\mathbf{X}}, \widetilde{\mathbf{A}}) = \mathcal{C}(\mathbf{X}, \mathbf{A})$. The choice of the negative sampling procedure will govern the specific kinds of structural information that is desirable to be captured as a byproduct of this maximization.

For the objective, we follow the intuitions from Deep InfoMax (DIM, Hjelm et al., 2018) and use a noise-contrastive type objective with a standard binary cross-entropy (BCE) loss between the samples from the joint (positive examples) and the product of marginals (negative examples). Following

their work, we use the following objective[1]:

$$\mathcal{L} = \frac{1}{N+M} \left( \sum_{i=1}^{N} \mathbb{E}_{(\mathbf{X}, \mathbf{A})} \left[ \log \mathcal{D} \left( \vec{h}_i, \vec{s} \right) \right] + \sum_{j=1}^{M} \mathbb{E}_{(\widetilde{\mathbf{X}}, \widetilde{\mathbf{A}})} \left[ \log \left( 1 - \mathcal{D} \left( \vec{\tilde{h}}_j, \vec{s} \right) \right) \right] \right) \quad (1)$$

This approach effectively maximizes mutual information between $\vec{h}_i$ and $\vec{s}$, based on the Jensen-Shannon divergence[2] between the joint and the product of marginals.

As all of the derived patch representations are driven to preserve mutual information with the global graph summary, this allows for discovering and preserving similarities on the patch-level—for example, distant nodes with similar structural roles (which are known to be a strong predictor for many node classification tasks; Donnat et al., 2018). Note that this is a "reversed" version of the argument given by Hjelm et al. (2018): for node classification, our aim is for the *patches* to establish links to similar patches across the graph, rather than enforcing the summary to contain all of these similarities (however, both of these effects should in principle occur simultaneously).

## 3.3 THEORETICAL MOTIVATION

We now provide some intuition that connects the classification error of our discriminator to mutual information maximization on graph representations.

**Lemma 1.** *Let $\{\mathbf{X}^{(k)}\}_{k=1}^{|\mathbf{X}|}$ be a set of node representations drawn from an empirical probability distribution of graphs, $p(\mathbf{X})$, with finite number of elements, $|\mathbf{X}|$, such that $p(\mathbf{X}^{(k)}) = p(\mathbf{X}^{(k')}) \, \forall k, k'$. Let $\mathcal{R}(\cdot)$ be a deterministic readout function on graphs and $\vec{s}^{(k)} = \mathcal{R}(\mathbf{X}^{(k)})$ be the summary vector of the $k$-th graph, with marginal distribution $p(\vec{s})$. The optimal classifier between the joint distribution $p(\mathbf{X}, \vec{s})$ and the product of marginals $p(\mathbf{X})p(\vec{s})$, assuming class balance, has an error rate upper bounded by $\mathrm{Err}^* = \frac{1}{2} \sum_{k=1}^{|\mathbf{X}|} p(\vec{s}^{(k)})^2$. This upper bound is achieved if $\mathcal{R}$ is injective.*

*Proof.* Denote by $\mathcal{Q}^{(k)}$ the set of all graphs in the input set that are mapped to $\vec{s}^{(k)}$ by $\mathcal{R}$, i.e. $\mathcal{Q}^{(k)} = \{\mathbf{X}^{(j)} \mid \mathcal{R}(\mathbf{X}^{(j)}) = \vec{s}^{(k)}\}$. As $\mathcal{R}(\cdot)$ is deterministic, samples from the joint, $(\mathbf{X}^{(k)}, \vec{s}^{(k)})$ are drawn from the product of marginals with probability $p(\vec{s}^{(k)})p(\mathbf{X}^{(k)})$, which decomposes into:

$$p(\vec{s}^{(k)}) \sum_{\vec{s}} p(\mathbf{X}^{(k)}, \vec{s}) = p(\vec{s}^{(k)}) p(\mathbf{X}^{(k)} | \vec{s}^{(k)}) p(\vec{s}^{(k)}) = \frac{p(\mathbf{X}^{(k)})}{\sum_{\mathbf{X}' \in \mathcal{Q}^{(k)}} p(\mathbf{X}')} p(\vec{s}^{(k)})^2 \quad (2)$$

For convenience, let $\rho^{(k)} = \frac{p(\mathbf{X}^{(k)})}{\sum_{\mathbf{X}' \in \mathcal{Q}^{(k)}} p(\mathbf{X}')}$. As, by definition, $\mathbf{X}^{(k)} \in \mathcal{Q}^{(k)}$, it holds that $\rho^{(k)} \leq 1$. This probability ratio is maximized at 1 when $\mathcal{Q}^{(k)} = \{\mathbf{X}^{(k)}\}$, i.e. when $\mathcal{R}$ is injective for $\mathbf{X}^{(k)}$. The probability of drawing any sample of the joint from the product of marginals is then bounded above by $\sum_{k=1}^{|\mathbf{X}|} p(\vec{s}^{(k)})^2$. As the probability of drawing $(\mathbf{X}^{(k)}, \vec{s}^{(k)})$ from the joint is $\rho^{(k)} p(\vec{s}^{(k)}) \geq \rho^{(k)} p(\vec{s}^{(k)})^2$, we know that classifying these samples as coming from the joint has a lower error than classifying them as coming from the product of marginals. The error rate of such a classifier is then the probability of drawing a sample from the joint as a sample from product of marginals under the mixture probability, which we can bound by $\mathrm{Err} \leq \frac{1}{2} \sum_{k=1}^{|\mathbf{X}|} p(\vec{s}^{(k)})^2$, with the upper bound achieved, as above, when $\mathcal{R}(\cdot)$ is injective for all elements of $\{\mathbf{X}^{(k)}\}$. □

It may be useful to note that $\frac{1}{2|\mathbf{X}|} \leq \mathrm{Err}^* \leq \frac{1}{2}$. The first result is obtained via a trivial application of Jensen's inequality, while the other extreme is reached only in the edge case of a constant readout function (when every example from the joint is also an example from the product of marginals, so no classifier performs better than chance).

**Corollary 1.** *From now on, assume that the readout function used, $\mathcal{R}$, is injective. Assume the number of allowable states in the space of $\vec{s}$, $|\vec{s}|$, is greater than or equal to $|\mathbf{X}|$. Then, for $\vec{s}^\star$, the*

---

[1]Note that Hjelm et al. (2018) use a softplus version of the binary cross-entropy.

[2]The "GAN" distance defined here—as per Goodfellow et al. (2014) and Nowozin et al. (2016)—and Jensen-Shannon divergence can be related by $D_{GAN} = 2D_{JS} - \log 4$. Therefore, any parameters that optimize one also optimize the other.

*optimal summary under the classification error of an optimal classifier between the joint and the product of marginals, it holds that $|\vec{s}^\star| = |\mathbf{X}|$.*

*Proof.* By injectivity of $\mathcal{R}$, we know that $\vec{s}^\star = \operatorname{argmin}_{\vec{s}} \operatorname{Err}^*$. As the upper error bound, $\operatorname{Err}^*$, is a simple geometric sum, we know that this is minimized when $p(\vec{s}^{(k)})$ is uniform. As $\mathcal{R}(\cdot)$ is deterministic, this implies that each potential summary state would need to be used at least once. Combined with the condition $|\vec{s}| \geq |\mathbf{X}|$, we conclude that the optimum has $|\vec{s}^\star| = |\mathbf{X}|$. □

**Theorem 1.** $\vec{s}^\star = \operatorname{argmax}_{\vec{s}} \operatorname{MI}(\mathbf{X}; \vec{s})$*, where* $\operatorname{MI}$ *is mutual information.*

*Proof.* This follows from the fact that the mutual information is invariant under invertible transforms. As $|\vec{s}^\star| = |\mathbf{X}|$ and $\mathcal{R}$ is injective, it has an inverse function, $\mathcal{R}^{-1}$. It follows then that, for any $\vec{s}$, $\operatorname{MI}(\mathbf{X}; \vec{s}) \leq H(\mathbf{X}) = \operatorname{MI}(\mathbf{X}; \mathbf{X}) = \operatorname{MI}(\mathbf{X}; \mathcal{R}(\mathbf{X})) = \operatorname{MI}(\mathbf{X}; \vec{s}^\star)$, where $H$ is entropy. □

Theorem 1 shows that for finite input sets and suitable deterministic functions, minimizing the classification error in the discriminator can be used to maximize the mutual information between the input and output. However, as was shown in Hjelm et al. (2018), this objective alone is not enough to learn useful representations. As in their work, we discriminate between the global summary vector and local high-level representations.

**Theorem 2.** *Let* $\mathbf{X}_i^{(k)} = \{\vec{x}_j\}_{j \in n(\mathbf{X}^{(k)}, i)}$ *be the neighborhood of the node* $i$ *in the* $k$*-th graph that collectively maps to its high-level features,* $\vec{h}_i = \mathcal{E}(\mathbf{X}_i^{(k)})$*, where* $n$ *is the neighborhood function that returns the set of neighborhood indices of node* $i$ *for graph* $\mathbf{X}^{(k)}$*, and* $\mathcal{E}$ *is a deterministic encoder function. Let us assume that* $|\mathbf{X}_i| = |\mathbf{X}| = |\vec{s}| \geq |\vec{h}_i|$*. Then, the* $\vec{h}_i$ *that minimizes the classification error between* $p(\vec{h}_i, \vec{s})$ *and* $p(\vec{h}_i)p(\vec{s})$ *also maximizes* $\operatorname{MI}(\mathbf{X}_i^{(k)}; \vec{h}_i)$*.*

*Proof.* Given our assumption of $|\mathbf{X}_i| = |\vec{s}|$, there exists an inverse $\mathbf{X}_i = \mathcal{R}^{-1}(\vec{s})$, and therefore $\vec{h}_i = \mathcal{E}(\mathcal{R}^{-1}(\vec{s}))$, i.e. there exists a deterministic function $(\mathcal{E} \circ \mathcal{R}^{-1})$ mapping $\vec{s}$ to $\vec{h}_i$. The optimal classifier between the joint $p(\vec{h}_i, \vec{s})$ and the product of marginals $p(\vec{h}_i)p(\vec{s})$ then has (by Lemma 1) an error rate upper bound of $\operatorname{Err}^* = \frac{1}{2} \sum_{k=1}^{|\mathbf{X}|} p(\vec{h}_i^{(k)})^2$. Therefore (as in Corollary 1), for the optimal $\vec{h}_i$, $|\vec{h}_i| = |\mathbf{X}_i|$, which by the same arguments as in Theorem 1 maximizes the mutual information between the neighborhood and high-level features, $\operatorname{MI}(\mathbf{X}_i^{(k)}; \vec{h}_i)$. □

This motivates our use of a classifier between samples from the joint and the product of marginals, and using the binary cross-entropy (BCE) loss to optimize this classifier is well-understood in the context of neural network optimization.

### 3.4 OVERVIEW OF DGI

Assuming the single-graph setup (i.e., $(\mathbf{X}, \mathbf{A})$ provided as input), we will now summarize the steps of the Deep Graph Infomax procedure:

1. Sample a negative example by using the corruption function: $(\widetilde{\mathbf{X}}, \widetilde{\mathbf{A}}) \sim \mathcal{C}(\mathbf{X}, \mathbf{A})$.

2. Obtain patch representations, $\vec{h}_i$ for the input graph by passing it through the encoder: $\mathbf{H} = \mathcal{E}(\mathbf{X}, \mathbf{A}) = \{\vec{h}_1, \vec{h}_2, \dots, \vec{h}_N\}$.

3. Obtain patch representations, $\vec{\tilde{h}}_j$ for the negative example by passing it through the encoder: $\widetilde{\mathbf{H}} = \mathcal{E}(\widetilde{\mathbf{X}}, \widetilde{\mathbf{A}}) = \{\vec{\tilde{h}}_1, \vec{\tilde{h}}_2, \dots, \vec{\tilde{h}}_M\}$.

4. Summarize the input graph by passing its patch representations through the readout function: $\vec{s} = \mathcal{R}(\mathbf{H})$.

5. Update parameters of $\mathcal{E}$, $\mathcal{R}$ and $\mathcal{D}$ by applying gradient descent to maximize Equation 1.

This algorithm is fully summarized by Figure 1.

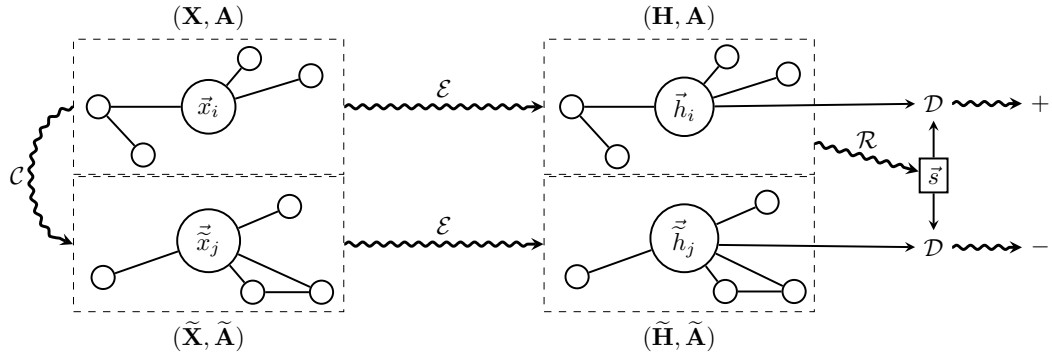

Figure 1: A high-level overview of Deep Graph Infomax. Refer to Section 3.4 for more details.

Table 1: Summary of the datasets used in our experiments.

| Dataset | Task | Nodes | Edges | Features | Classes | Train/Val/Test Nodes |
|---|---|---|---|---|---|---|
| **Cora** | Transductive | 2,708 | 5,429 | 1,433 | 7 | 140/500/1,000 |
| **Citeseer** | Transductive | 3,327 | 4,732 | 3,703 | 6 | 120/500/1,000 |
| **Pubmed** | Transductive | 19,717 | 44,338 | 500 | 3 | 60/500/1,000 |
| **Reddit** | Inductive | 231,443 | 11,606,919 | 602 | 41 | 151,708/23,699/55,334 |
| **PPI** | Inductive | 56,944 (24 graphs) | 818,716 | 50 | 121 (multilbl.) | 44,906/6,514/5,524 (20/2/2 graphs) |

## 4 CLASSIFICATION PERFORMANCE

We have assessed the benefits of the representation learnt by the DGI encoder on a variety of node classification tasks (transductive as well as inductive), obtaining competitive results. In each case, DGI was used to learn patch representations in a fully unsupervised manner, followed by evaluating the node-level classification utility of these representations. This was performed by directly using these representations to train and test a simple linear (logistic regression) classifier.

### 4.1 DATASETS

We follow the experimental setup described in Kipf & Welling (2016a) and Hamilton et al. (2017a) on the following benchmark tasks: (1) classifying research papers into topics on the Cora, Citeseer and Pubmed citation networks (Sen et al., 2008); (2) predicting the community structure of a social network modeled with Reddit posts; and (3) classifying protein roles within protein-protein interaction (PPI) networks (Zitnik & Leskovec, 2017), requiring generalisation to unseen networks.

Further information on the datasets may be found in Table 1 and Appendix A.

### 4.2 EXPERIMENTAL SETUP

For each of three experimental settings (transductive learning, inductive learning on large graphs, and multiple graphs), we employed distinct encoders and corruption functions appropriate to that setting (described below).

**Transductive learning**. For the transductive learning tasks (Cora, Citeseer and Pubmed), our encoder is a one-layer Graph Convolutional Network (GCN) model (Kipf & Welling, 2016a), with the following propagation rule:

$$\mathcal{E}(\mathbf{X}, \mathbf{A}) = \sigma \left( \hat{\mathbf{D}}^{-\frac{1}{2}} \hat{\mathbf{A}} \hat{\mathbf{D}}^{-\frac{1}{2}} \mathbf{X} \mathbf{\Theta} \right) \tag{3}$$

where $\hat{\mathbf{A}} = \mathbf{A} + \mathbf{I}_N$ is the adjacency matrix with inserted self-loops and $\hat{\mathbf{D}}$ is its corresponding degree matrix; i.e. $\hat{D}_{ii} = \sum_j \hat{A}_{ij}$. For the nonlinearity, $\sigma$, we have applied the parametric ReLU

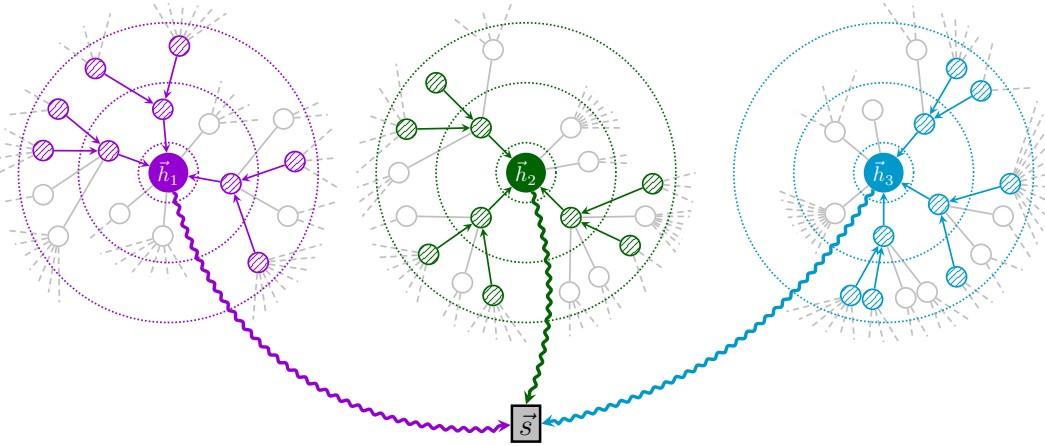

Figure 2: The DGI setup on large graphs (such as Reddit). Summary vectors, $\vec{s}$, are obtained by combining several subsampled patch representations, $\vec{h}_i$ (here obtained by sampling three and two neighbors in the first and second level, respectively).

(PReLU) function (He et al., 2015), and $\boldsymbol{\Theta} \in \mathbb{R}^{F \times F'}$ is a learnable linear transformation applied to every node, with $F' = 512$ features being computed (specially, $F' = 256$ on Pubmed due to memory limitations).

The corruption function used in this setting is designed to encourage the representations to properly encode structural similarities of different nodes in the graph; for this purpose, $\mathcal{C}$ preserves the original adjacency matrix ($\widetilde{\mathbf{A}} = \mathbf{A}$), whereas the corrupted features, $\widetilde{\mathbf{X}}$, are obtained by row-wise shuffling of $\mathbf{X}$. That is, the corrupted graph consists of exactly the same nodes as the original graph, but they are located in different places in the graph, and will therefore receive different patch representations. We demonstrate DGI is stable to other choices of corruption functions in Appendix C, but we find those that preserve the graph structure result in the strongest features.

**Inductive learning on large graphs**. For inductive learning, we may no longer use the GCN update rule in our encoder (as the learned filters rely on a fixed and known adjacency matrix); instead, we apply the *mean-pooling* propagation rule, as used by GraphSAGE-GCN (Hamilton et al., 2017a):

$$\text{MP}(\mathbf{X}, \mathbf{A}) = \hat{\mathbf{D}}^{-1}\hat{\mathbf{A}}\mathbf{X}\boldsymbol{\Theta} \tag{4}$$

with parameters defined as in Equation 3. Note that multiplying by $\hat{\mathbf{D}}^{-1}$ actually performs a normalized sum (hence the mean-pooling). While Equation 4 explicitly specifies the adjacency and degree matrices, *they are not needed*: identical inductive behaviour may be observed by a *constant* attention mechanism across the node's neighbors, as used by the Const-GAT model (Veličković et al., 2018).

For Reddit, our encoder is a three-layer mean-pooling model with skip connections (He et al., 2016):

$$\widetilde{\text{MP}}(\mathbf{X}, \mathbf{A}) = \sigma\left(\mathbf{X}\boldsymbol{\Theta}' \| \text{MP}(\mathbf{X}, \mathbf{A})\right) \qquad \mathcal{E}(\mathbf{X}, \mathbf{A}) = \widetilde{\text{MP}}_3(\widetilde{\text{MP}}_2(\widetilde{\text{MP}}_1(\mathbf{X}, \mathbf{A}), \mathbf{A}), \mathbf{A}) \tag{5}$$

where $\|$ is featurewise concatenation (i.e. the central node and its neighborhood are handled separately). We compute $F' = 512$ features in each MP layer, with the PReLU activation for $\sigma$.

Given the large scale of the dataset, it will not fit into GPU memory entirely. Therefore, we use the subsampling approach of Hamilton et al. (2017a), where a minibatch of nodes is first selected, and then a subgraph centered around each of them is obtained by *sampling node neighborhoods with replacement*. Specifically, we sample 10, 10 and 25 neighbors at the first, second and third level, respectively—thus, each subsampled patch has 1 + 10 + 100 + 2500 = 2611 nodes. Only the computations necessary for deriving the central node $i$'s patch representation, $\vec{h}_i$, are performed. These representations are then used to derive the summary vector, $\vec{s}$, for the minibatch (Figure 2). We used minibatches of 256 nodes throughout training.

To define our corruption function in this setting, we use a similar approach as in the transductive tasks, but treat each subsampled patch as a separate graph to be corrupted (i.e., we row-wise shuffle

the feature matrices within a subsampled patch). Note that this may very likely cause the central node's features to be swapped out for a sampled neighbor's features, further encouraging diversity in the negative samples. The patch representation obtained in the central node is then submitted to the discriminator.

**Inductive learning on multiple graphs**. For the PPI dataset, inspired by previous successful supervised architectures (Veličković et al., 2018), our encoder is a three-layer mean-pooling model with dense skip connections (He et al., 2016; Huang et al., 2017):

$$\mathbf{H}_1 = \sigma\left(\mathrm{MP}_1(\mathbf{X}, \mathbf{A})\right) \tag{6}$$

$$\mathbf{H}_2 = \sigma\left(\mathrm{MP}_2(\mathbf{H}_1 + \mathbf{X}\mathbf{W}_{\mathrm{skip}}, \mathbf{A})\right) \tag{7}$$

$$\mathcal{E}(\mathbf{X}, \mathbf{A}) = \sigma\left(\mathrm{MP}_3(\mathbf{H}_2 + \mathbf{H}_1 + \mathbf{X}\mathbf{W}_{\mathrm{skip}}, \mathbf{A})\right) \tag{8}$$

where $\mathbf{W}_{\mathrm{skip}}$ is a learnable projection matrix, and MP is as defined in Equation 4. We compute $F' = 512$ features in each MP layer, using the PReLU activation for $\sigma$.

In this multiple-graph setting, we opted to use *randomly sampled training graphs* as negative examples (i.e., our corruption function simply samples a different graph from the training set). We found this method to be the most stable, considering that over 40% of the nodes have all-zero features in this dataset. To further expand the pool of negative examples, we also apply dropout (Srivastava et al., 2014) to the input features of the sampled graph. We found it beneficial to standardize the learnt embeddings across the training set prior to providing them to the logistic regression model.

**Readout, discriminator, and additional training details**. Across all three experimental settings, we employed identical readout functions and discriminator architectures.

For the readout function, we use a simple averaging of all the nodes' features:

$$\mathcal{R}(\mathbf{H}) = \sigma\left(\frac{1}{N}\sum_{i=1}^{N}\vec{h}_i\right) \tag{9}$$

where $\sigma$ is the logistic sigmoid nonlinearity. While we have found this readout to perform the best across all our experiments, we assume that its power will diminish with the increase in graph size, and in those cases, more sophisticated readout architectures such as set2vec (Vinyals et al., 2015) or DiffPool (Ying et al., 2018b) are likely to be more appropriate.

The discriminator scores summary-patch representation pairs by applying a simple bilinear scoring function (similar to the scoring used by Oord et al. (2018)):

$$\mathcal{D}(\vec{h}_i, \vec{s}) = \sigma\left(\vec{h}_i^T \mathbf{W} \vec{s}\right) \tag{10}$$

Here, $\mathbf{W}$ is a learnable scoring matrix and $\sigma$ is the logistic sigmoid nonlinearity, used to convert scores into probabilities of $(\vec{h}_i, \vec{s})$ being a positive example.

All models are initialized using Glorot initialization (Glorot & Bengio, 2010) and trained to maximize the mutual information provided in Equation 1 on the available nodes (all nodes for the transductive, and training nodes only in the inductive setup) using the Adam SGD optimizer (Kingma & Ba, 2014) with an initial learning rate of 0.001 (specially, $10^{-5}$ on Reddit). On the transductive datasets, we use an early stopping strategy on the observed *training* loss, with a patience of 20 epochs[3]. On the inductive datasets we train for a fixed number of epochs (150 on Reddit, 20 on PPI).

### 4.3  RESULTS

The results of our comparative evaluation experiments are summarized in Table 2.

For the transductive tasks, we report the mean classification accuracy (with standard deviation) on the test nodes of our method after 50 runs of training (followed by logistic regression), and reuse the metrics already reported in Kipf & Welling (2016a) for the performance of DeepWalk and GCN, as well as Label Propagation (LP) (Zhu et al., 2003) and Planetoid (Yang et al., 2016)—a representative supervised random walk method. Specially, we provide results for training the logistic regression on raw input features, as well as DeepWalk with the input features concatenated.

---

[3] A reference DGI implementation may be found at `https://github.com/PetarV-/DGI`.

Table 2: Summary of results in terms of classification accuracies (on transductive tasks) or micro-averaged $F_1$ scores (on inductive tasks). In the first column, we highlight the kind of data available to each method during training ($\mathbf{X}$: features, $\mathbf{A}$: adjacency matrix, $\mathbf{Y}$: labels). "GCN" corresponds to a two-layer DGI encoder trained in a supervised manner.

*Transductive*

| Available data | Method | Cora | Citeseer | Pubmed |
|---|---|---|---|---|
| $\mathbf{X}$ | Raw features | $47.9 \pm 0.4\%$ | $49.3 \pm 0.2\%$ | $69.1 \pm 0.3\%$ |
| $\mathbf{A}, \mathbf{Y}$ | LP (Zhu et al., 2003) | $68.0\%$ | $45.3\%$ | $63.0\%$ |
| $\mathbf{A}$ | DeepWalk (Perozzi et al., 2014) | $67.2\%$ | $43.2\%$ | $65.3\%$ |
| $\mathbf{X}, \mathbf{A}$ | DeepWalk + features | $70.7 \pm 0.6\%$ | $51.4 \pm 0.5\%$ | $74.3 \pm 0.9\%$ |
| $\mathbf{X}, \mathbf{A}$ | Random-Init (ours) | $69.3 \pm 1.4\%$ | $61.9 \pm 1.6\%$ | $69.6 \pm 1.9\%$ |
| $\mathbf{X}, \mathbf{A}$ | **DGI** (ours) | $\mathbf{82.3} \pm 0.6\%$ | $\mathbf{71.8} \pm 0.7\%$ | $\mathbf{76.8} \pm 0.6\%$ |
| $\mathbf{X}, \mathbf{A}, \mathbf{Y}$ | GCN (Kipf & Welling, 2016a) | $81.5\%$ | $70.3\%$ | $79.0\%$ |
| $\mathbf{X}, \mathbf{A}, \mathbf{Y}$ | Planetoid (Yang et al., 2016) | $75.7\%$ | $64.7\%$ | $77.2\%$ |

*Inductive*

| Available data | Method | Reddit | PPI |
|---|---|---|---|
| $\mathbf{X}$ | Raw features | 0.585 | 0.422 |
| $\mathbf{A}$ | DeepWalk (Perozzi et al., 2014) | 0.324 | — |
| $\mathbf{X}, \mathbf{A}$ | DeepWalk + features | 0.691 | — |
| $\mathbf{X}, \mathbf{A}$ | GraphSAGE-GCN (Hamilton et al., 2017a) | 0.908 | 0.465 |
| $\mathbf{X}, \mathbf{A}$ | GraphSAGE-mean (Hamilton et al., 2017a) | 0.897 | 0.486 |
| $\mathbf{X}, \mathbf{A}$ | GraphSAGE-LSTM (Hamilton et al., 2017a) | 0.907 | 0.482 |
| $\mathbf{X}, \mathbf{A}$ | GraphSAGE-pool (Hamilton et al., 2017a) | 0.892 | 0.502 |
| $\mathbf{X}, \mathbf{A}$ | Random-Init (ours) | $0.933 \pm 0.001$ | $0.626 \pm 0.002$ |
| $\mathbf{X}, \mathbf{A}$ | **DGI** (ours) | $\mathbf{0.940} \pm 0.001$ | $\mathbf{0.638} \pm 0.002$ |
| $\mathbf{X}, \mathbf{A}, \mathbf{Y}$ | FastGCN (Chen et al., 2018) | 0.937 | — |
| $\mathbf{X}, \mathbf{A}, \mathbf{Y}$ | Avg. pooling (Zhang et al., 2018) | $0.958 \pm 0.001$ | $0.969 \pm 0.002$ |

For the inductive tasks, we report the micro-averaged $F_1$ score on the (unseen) test nodes, averaged after 50 runs of training, and reuse the metrics already reported in Hamilton et al. (2017a) for the other techniques. Specifically, as our setup is unsupervised, we compare against the unsupervised GraphSAGE approaches. We also provide supervised results for two related architectures—FastGCN (Chen et al., 2018) and Avg. pooling (Zhang et al., 2018).

Our results demonstrate strong performance being achieved across all five datasets. We particularly note that the DGI approach is competitive with the results reported for the GCN model *with the supervised loss*, even exceeding its performance on the Cora and Citeseer datasets. We assume that these benefits stem from the fact that, indirectly, the DGI approach allows for every node to have access to structural properties of the entire graph, whereas the supervised GCN is limited to only two-layer neighborhoods (by the extreme sparsity of the training signal and the corresponding threat of overfitting). It should be noted that, while we are capable of outperforming equivalent supervised encoder architectures, our performance still does not surpass the current supervised transductive state of the art (which is held by methods such as GraphSGAN (Ding et al., 2018)). We further observe that the DGI method successfully outperformed all the competing unsupervised GraphSAGE approaches on the Reddit and PPI datasets—thus verifying the potential of methods based on local mutual information maximization in the inductive node classification domain. Our Reddit results are competitive with the supervised state of the art, whereas on PPI the gap is still large—we believe this can be attributed to the extreme sparsity of available node features (over 40% of the nodes having all-zero features), that our encoder heavily relies on.

We note that a *randomly initialized* graph convolutional network may already extract highly useful features and represents a strong baseline—a well-known fact, considering its links to the Weisfeiler-

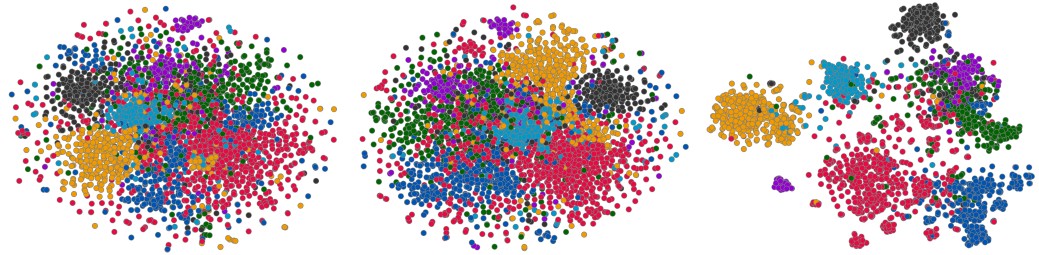

Figure 3: t-SNE embeddings of the nodes in the Cora dataset from the raw features (**left**), features from a randomly initialized DGI model (**middle**), and a learned DGI model (**right**). The clusters of the learned DGI model's embeddings are clearly defined, with a Silhouette score of 0.234.

Lehman graph isomorphism test (Weisfeiler & Lehman, 1968), that have already been highlighted and analyzed by Kipf & Welling (2016a) and Hamilton et al. (2017a). As such, we also provide, as *Random-Init*, the logistic regression performance on embeddings obtained from a randomly initialized encoder. Besides demonstrating that DGI is able to further improve on this strong baseline, it particularly reveals that, on the inductive datasets, previous random walk-based negative sampling methods may have been ineffective for learning appropriate features for the classification task.

Lastly, it should be noted that deeper encoders correspond to more pronounced *mixing* between recovered patch representations, reducing the effective variability of our positive/negative examples' pool. We believe that this is the reason why shallower architectures performed better on some of the datasets. While we cannot say that these trends will hold in general, with the DGI loss function we generally found benefits from employing *wider*, rather than *deeper* models.

## 5 Qualitative analysis

We performed a diverse set of analyses on the embeddings learnt by the DGI algorithm in order to better understand the properties of DGI. We focus our analysis exclusively on the Cora dataset (as it has the smallest number of nodes, significantly aiding clarity).

A standard set of "evolving" t-SNE plots (Maaten & Hinton, 2008) of the embeddings is given in Figure 3. As expected given the quantitative results, the learnt embeddings' 2D projections exhibit discernible clustering in the 2D projected space (especially compared to the raw features and Random-Init), which respects the seven topic classes of Cora. The projection obtains a Silhouette score (Rousseeuw, 1987) of 0.234, which compares favorably with the previous reported score of 0.158 for Embedding Propagation (Duran & Niepert, 2017).

We ran further analyses, revealing insights into DGI's mechanism of learning, isolating *biased* embedding dimensions for pushing the negative example scores down and using the remainder to encode useful information about positive examples. We leverage these insights to retain competitive performance to the supervised GCN even after *half* the dimensions are removed from the patch representations provided by the encoder. These—and several other—qualitative and ablation studies can be found in Appendix B.

## 6 Conclusions

We have presented Deep Graph Infomax (DGI), a new approach for learning unsupervised representations on graph-structured data. By leveraging local mutual information maximization across the graph's patch representations, obtained by powerful graph convolutional architectures, we are able to obtain node embeddings that are mindful of the global structural properties of the graph. This enables competitive performance across a variety of both transductive and inductive classification tasks, at times even outperforming relevant *supervised* architectures.

ACKNOWLEDGMENTS

We would like to thank the developers of PyTorch (Paszke et al., 2017). PV and PL have received funding from the European Union's Horizon 2020 research and innovation programme PROPAG-AGEING under grant agreement No 634821. We specially thank Hugo Larochelle and Jian Tang for the extremely useful discussions, and Andreea Deac, Arantxa Casanova, Ben Poole, Graham Taylor, Guillem Cucurull, Justin Gilmer, Nithium Thain and Zhaocheng Zhu for reviewing the paper prior to submission.

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

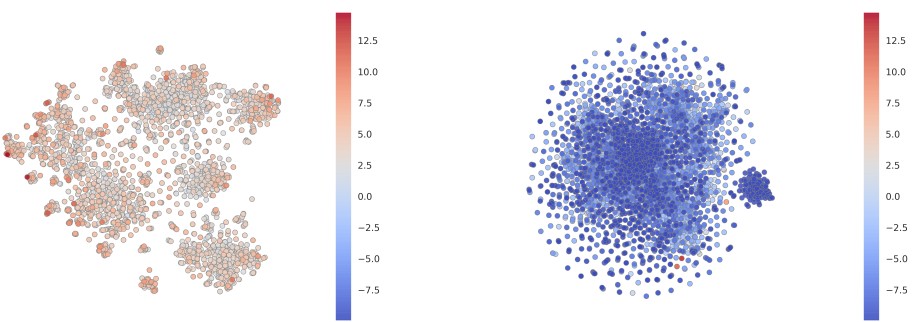

Figure 4: Discriminator scores, $\mathcal{D}\left(\vec{h}_i, \vec{s}\right)$, attributed to each node in the Cora dataset shown over a t-SNE of the DGI algorithm. Shown for both the original graph (**left**) and a negative sample (**right**).

## A  FURTHER DATASET DETAILS

**Transductive learning**. We utilize three standard citation network benchmark datasets—Cora, Citeseer and Pubmed (Sen et al., 2008)—and closely follow the transductive experimental setup of Yang et al. (2016). In all of these datasets, nodes correspond to documents and edges to (undirected) citations. Node features correspond to elements of a bag-of-words representation of a document. Each node has a class label. We allow for only 20 nodes per class to be used for training—however, honouring the transductive setup, the unsupervised learning algorithm has access to all of the nodes' feature vectors. The predictive power of the learned representations is evaluated on 1000 test nodes.

**Inductive learning on large graphs**. We use a large graph dataset (231,443 nodes and 11,606,919 edges) of Reddit posts created during September 2014 (derived and preprocessed as in Hamilton et al. (2017a)). The objective is to predict the posts' community (*"subreddit"*), based on the GloVe embeddings of their content and comments (Pennington et al., 2014), as well as metrics such as score or number of comments. Posts are linked together in the graph if the same user has commented on both. Reusing the inductive setup of Hamilton et al. (2017a), posts made in the first 20 days of the month are used for training, while the remaining posts are used for validation or testing and are *invisible* to the training algorithm.

**Inductive learning on multiple graphs**. We make use of a protein-protein interaction (PPI) dataset that consists of graphs corresponding to different human tissues (Zitnik & Leskovec, 2017). The dataset contains 20 graphs for training, 2 for validation and 2 for testing. Critically, testing graphs remain *completely unobserved* during training. To construct the graphs, we used the preprocessed data provided by Hamilton et al. (2017a). Each node has 50 features that are composed of positional gene sets, motif gene sets and immunological signatures. There are 121 labels for each node set from gene ontology, collected from the Molecular Signatures Database (Subramanian et al., 2005), and a node can possess several labels simultaneously.

## B  FURTHER QUALITATIVE ANALYSIS

**Visualizing discriminator scores**. After obtaining the t-SNE visualizations, we turned our attention to the discriminator—and visualized the scores it attached to various nodes, for both the positive and a (randomly sampled) negative example (Figure 4). From here we can make an interesting observation—within the "clusters" of the learnt embeddings on the positive Cora graph, only a handful of "hot" nodes are selected to receive high discriminator scores. This suggests that there may be a clear distinction between embedding dimensions used for discrimination and classification, which we more thoroughly investigate in the next paragraph. In addition, we may observe that, as expected, the model is unable to find any strong structure within a negative example. Lastly, a few negative examples achieve high discriminator scores—a phenomenon caused by the existence of

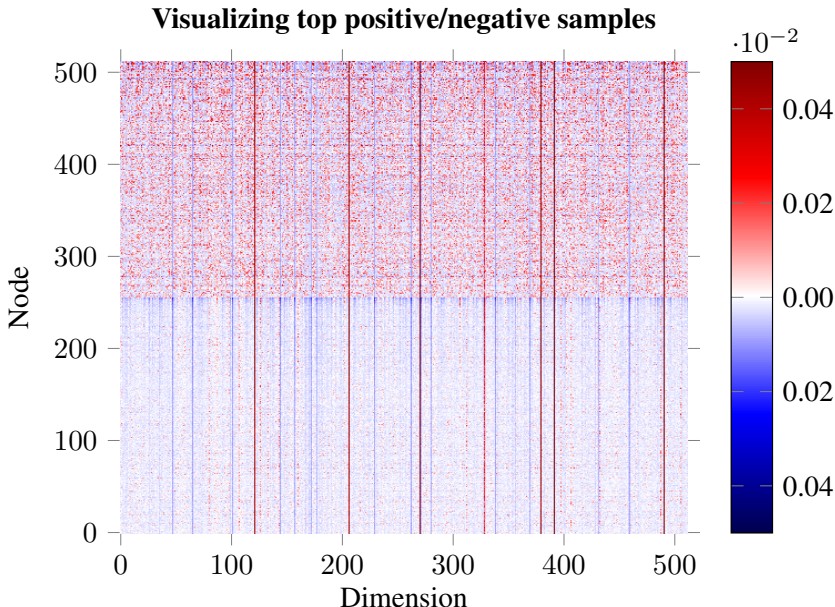

Figure 5: The learnt embeddings of the highest-scored positive examples (*upper half*), and the lowest-scored negative examples (*lower half*).

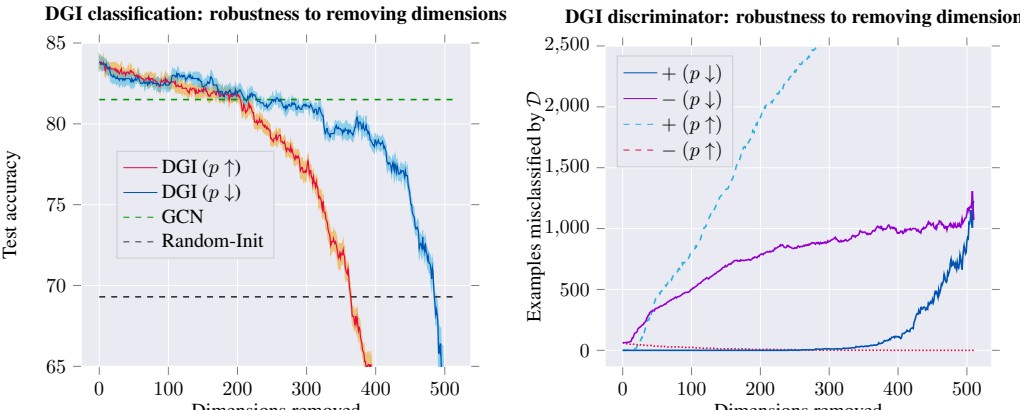

Figure 6: Classification performance (in terms of test accuracy of logistic regression; **left**) and discriminator performance (in terms of number of poorly discriminated positive/negative examples; **right**) on the learnt DGI embeddings, after removing a certain number of dimensions from the embedding—either starting with most distinguishing ($p \uparrow$) or least distinguishing ($p \downarrow$).

low-degree nodes in Cora (making the probability of a node ending up in an identical context it had in the positive graph non-negligible).

**Impact and role of embedding dimensions**. Guided by the previous result, we have visualized the embeddings for the top-scoring positive and negative examples (Figure 5). The analysis revealed existence of distinct dimensions in which both the positive and negative examples are *strongly biased*. We hypothesize that, given the random shuffling, the average *expected* activation of a negative example is zero, and therefore strong biases are required to "push" the example down in the discriminator. The positive examples may then use the remaining dimensions to both counteract this bias and encode patch similarity. To substantiate this claim, we order the 512 dimensions based on how distinguishable the positive and negative examples are in them (using $p$-values obtained from a t-test as a proxy). We then remove these dimensions from the embedding, respecting this order—either starting from the most distinguishable ($p \uparrow$) or least distinguishable dimensions ($p \downarrow$)—monitoring

how this affects both classification and discriminator performance (Figure 6). The observed trends largely support our hypothesis: if we start by removing the biased dimensions first ($p \downarrow$), the classification performance holds up for much longer (allowing us to remove over *half* of the embedding dimensions while remaining competitive to the supervised GCN), and the positive examples mostly remain correctly discriminated until well over half the dimensions are removed.

## C   ROBUSTNESS TO CHOICE OF CORRUPTION FUNCTION

Here, we consider alternatives to our corruption function, $\mathcal{C}$, used to produce negative graphs. We generally find that, for the node classification task, DGI is stable and robust to different strategies. However, for learning graph features towards other kinds of tasks, the design of appropriate corruption strategies remains an area of open research.

Our corruption function described in Section 4.2 preserves the original adjacency matrix ($\widetilde{\mathbf{A}} = \mathbf{A}$) but corrupts the features, $\widetilde{\mathbf{X}}$, via row-wise shuffling of $\mathbf{X}$. In this case, the negative graph is constrained to be isomorphic to the positive graph, which should not have to be mandatory. We can instead produce a negative graph by directly *corrupting* the adjacency matrix.

Therefore, we first consider an alternative corruption function $\mathcal{C}$ which preserves the features ($\widetilde{\mathbf{X}} = \mathbf{X}$) but instead adds or removes edges from the adjacency matrix ($\widetilde{\mathbf{A}} \neq \mathbf{A}$). This is done by sampling, i.i.d., a *switch* parameter $\mathbf{\Sigma}_{ij}$, which determines whether to corrupt the adjacency matrix at position $(i, j)$. Assuming a given *corruption rate*, $\rho$, we may define $\mathcal{C}$ as performing the following operations:

$$\mathbf{\Sigma}_{ij} \sim \text{Bernoulli}(\rho) \tag{11}$$

$$\widetilde{\mathbf{A}} = \mathbf{A} \oplus \mathbf{\Sigma} \tag{12}$$

where $\oplus$ is the XOR (exclusive OR) operation.

This alternative strategy produces a negative graph with the same features, but different connectivity. Here, the corruption rate of $\rho = 0$ corresponds to an unchanged adjacency matrix (i.e. the positive and negative graphs are *identical* in this case). In this regime, learning is impossible for the discriminator, and the performance of DGI is in line with a randomly initialized DGI model. At higher rates of noise, however, DGI produces competitive embeddings.

We also consider *simultaneous* feature shuffling ($\widetilde{\mathbf{X}} \neq \mathbf{X}$) and adjacency matrix perturbation ($\widetilde{\mathbf{A}} \neq \mathbf{A}$), both as described before. We find that DGI still learns useful features under this compound corruption strategy—as expected, given that feature shuffling is already equivalent to an (isomorphic) adjacency matrix perturbation.

From both studies, we may observe that a certain lower bound on the positive graph perturbation rate is required to obtain competitive node embeddings for the classification task on Cora. Furthermore, the features learned for downstream node classification tasks are most powerful when the negative graph has similar levels of connectivity to the positive graph.

The classification performance peaks when the graph is perturbed to a reasonably high level, but remains *sparse*; i.e. the mixing between the separate 1-step patches is not substantial, and therefore the pool of negative examples is still *diverse* enough. Classification performance is impacted only marginally at higher rates of corruption—corresponding to *dense* negative graphs, and thus a less rich negative example pool—but still considerably outperforming the unsupervised baselines we have considered. This could be seen as further motivation for relying solely on feature shuffling, without adjacency perturbations—given that feature shuffling is a trivial way to guarantee a diverse set of negative examples, without incurring significant computational costs per epoch.

The results of this study are visualized in Figures 7 and 8.

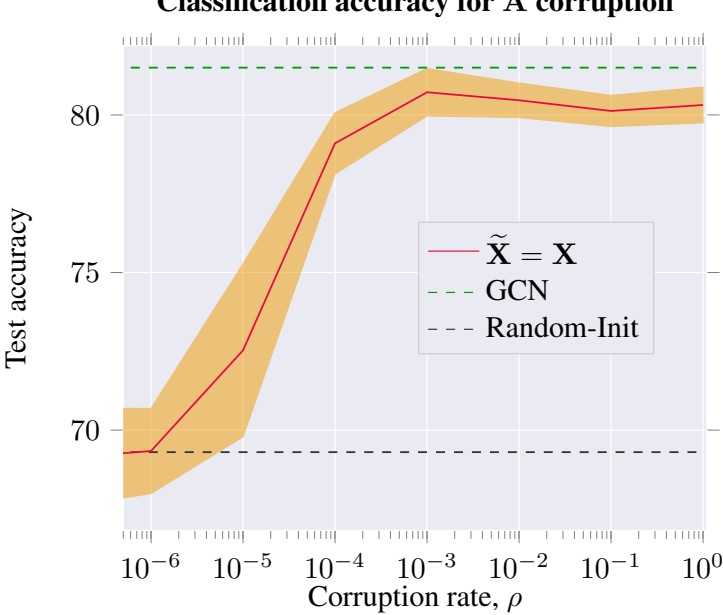

Figure 7: DGI also works under a corruption function that modifies only the adjacency matrix ($\widetilde{\mathbf{A}} \neq \mathbf{A}$) on the Cora dataset. The left range ($\rho \to 0$) corresponds to no modifications of the adjacency matrix—therein, performance approaches that of the randomly initialized DGI model. As $\rho$ increases, DGI produces more useful features, but ultimately fails to outperform the feature-shuffling corruption function. **N.B.** log scale used for $\rho$.

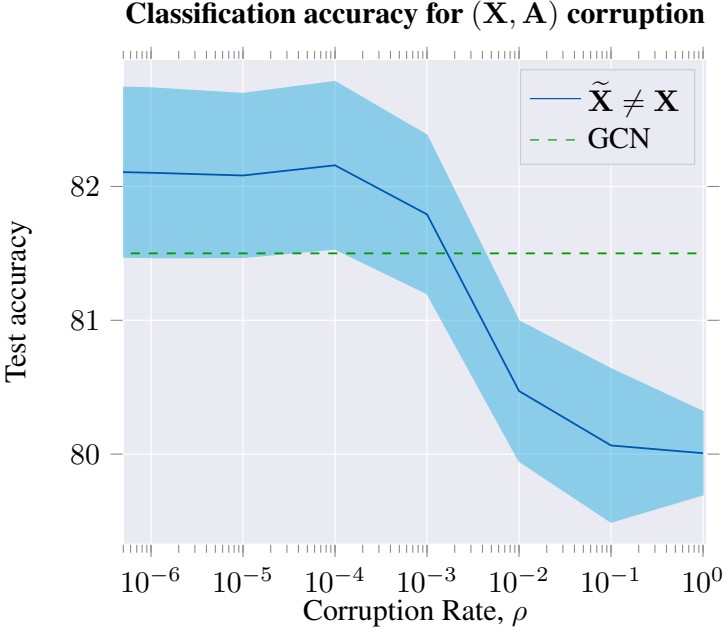

Figure 8: DGI is stable and robust under a corruption function that modifies *both* the feature matrix ($\mathbf{X} \neq \widetilde{\mathbf{X}}$) and the adjacency matrix ($\widetilde{\mathbf{A}} \neq \mathbf{A}$) on the Cora dataset. Corruption functions that preserve sparsity ($\rho \approx \frac{1}{N}$) perform the best. However, DGI still performs well even with large disruptions (where edges are added or removed with probabilities approaching 1). **N.B.** log scale used for $\rho$.

