# OpenReview forum: "Deep Graph Infomax"
_ICLR.cc/2019/Conference_

### Official Review · AnonReviewer2 · 2018-10-28
**Alternative information-theoretic objective for unsupervised graph representation learning**

**Rating:** 7
**Confidence:** 3

**Review:**

This paper adapts the Deep Informax (DIM; Hjelm et al. 2018) method, which was used on
image data, into the graph domain. The architecture of the neural network and
the learning cost function are given by figure 1 and eq.(1), respectively.

The idea is to maximize the mutual information between a local representation
(of a "patch" defined by graph adjacency) and a global representation (of the entire graph),
so those different local patches are encouraged to carry some shared
global information.

This is in contrast to most unsupervised graph encoders, where the objective is
to fit the random walk similarities (node adjacency on the graph).

In an unsupervised learning scenario, where the graph structure and node features
are given, the authors achieved state-of-the-art performance on transductive and
inductive node classification tasks, in some cases even better than supervised baselines.

The paper is well written. I recommend acceptance and have the following concerns.

Main comment 1

The title suggests that there are some information theory contents.
However, section 3 does not include much information theory.
Rather, the author(s) directly give eq.(1) with pointers to references and informal discussions.
This is not so helpful. It is not straightforward for
the reader to relate eq.(1) with the definition of mutual information.
Ideally, before eq.(1) there should be one or two equations (with text)
to introduce the Jesen-Shannon MI estimation and information theoretic bounds etc.

Overall, due to this, the contribution is mainly on adapting the DIM method info the graph domain. Although the experimental results are good, there is not much theoretical insight or "recreative" introduction of the DIM method from the authors' perspectives. This is the main reason for that it is not a strong accept.

Main comment 2

A motivation of the proposition is to "not rely on random walks", or graph node adjacency.
Notice that random walks can be intuitively regarded as higher order node adjacency.
However, the encoder, which is based on GCN, does rely on the adjacency matrix,
as the convolution is done in local neighborhoods (that can also be defined based on
random-walk similarities). The authors are therefore suggested to make it
clear in related places that, it is the cost function which is not based
on node adjacency, although the neural network structure does rely on it.

As a related question, in the inductive experiments, in the mini-batch of 256 nodes
randomly selected, or selected by a local patch of the graph which is connected or nearby?
If it is the latter case, the cost function does rely on random-walk similarities,
as the summary vector will be a local patch average.

Questions:

-The summary vector is the average of all node features. On large graphs, the
average may carry less information as compared to small graphs. It can be
observed that on Pubmed and Reddit, the performance improvement is not as
high as the other small graphs. Could you comment on this?

-In the baseline "DeepWalk+features", are the two different types of features directly concatenated?

-Is it straightforward to apply DGI to link prediction tasks?

-It that a concern that the random corruption function will cause a high variance of the gradient?

---

> ### Author Response · Authors · 2018-11-20
> **Reply to AnonReviewer2**
>
> Thank you for the very careful review and kind words about our contributions.
>
> Regarding your comment about our information-theoretic contributions, please see our global comment on mutual information and the JSD. We reorganised Sections 3.2 / 3.3 to include more concrete theoretical motivation from mutual information maximisation, and separate this from the specifics that are important for implementation.
>
> You make an exceptionally good point regarding our method’s reliance on random walks - thanks! It is, in fact, our main claim that combining random-walk *objectives* with GCN-like encoders is potentially unsuitable (given that the GCN already encodes the “random-walk” information within its structural inductive biases). We have appropriately modified our abstract to reflect this intention.
>
> To answer your remaining questions:
>
> - The minibatch of 256 nodes for Reddit is randomly selected, and therefore the cost function does not rely on random-walk similarities in this case.
>
> - Regarding the averaging readout: this is a great point, and we expect that the performance on larger graphs will decrease somewhat, especially when using averaging as a readout function, since it is known that the quality of graph-level embeddings degrades as the number of nodes increases when using simple averaging. That said, we expect this problem could be alleviated by applying more sophisticated set2vec and/or pooling approaches for the readout function, and we mention this point in the revised paper---immediately after the averaging is introduced in Section 4.2. Moreover, in this particular case, the smaller performance improvement on the Reddit data is also simply due to the fact that most GCN approaches are already in the 90+% F1 range, and therefore there is limited room for improvement.
>
> - The “DeepWalk+features” baseline directly concatenates the two kinds of features, as was done in all prior work utilising this baseline (e.g. Hamilton et al., NIPS 2017).
>
> - We note that we specifically designed the model with node classification tasks in mind. However, in principle, the generated embeddings could be used for link prediction, as with node2vec embeddings, etc. That said, we expect that strong performance on link prediction could require minor modifications (e.g., tweaking the negative sampling function) and we plan to investigate this in future work.
>
> - Regarding your concern about high variance of the gradient, we haven’t found any issues regarding learning stability---as long as an appropriate choice of learning rate is made.
>
> We thank you once again for your review, which has definitely helped make our paper’s contributions stronger!

---

> > ### Comment · AnonReviewer2 · 2018-11-26
> > **Comments after Revision**
> >
> > Thank you very much for the revision that has improved the paper. Kindly see below some comments:
> >
> > - Lemma 1
> > Use Jesnsen inequality and convexity of x^2 to have some trivial bounds of your bound (especially that it is smaller than 1).
> >
> > - Lemma 1, please double check if it is an upper bound or lower bound of the error rate
> >
> > - the stated theorems in section 3.3 is a bit weak because it requires the summarization to be invertible. Could you check whether similar results hold if the summarization vector is a sufficient statistics of X?
> >
> > - Conclusion, use comma instead of dash (or leave some spaces between dash and words)

---

> > > ### Author Response · Authors · 2018-11-30
> > > **Response to Comments**
> > >
> > > Thank you for following the discussion and for your further thoughts.
> > >
> > > As the square function is convex, Jensen's inequality will only give us a lower bound (of 1/2|X|) rather than an upper one. However, we can upper bound the sum of squares of probabilities to 1 without using Jensen's inequality (it's maximised for the one-hot distribution, as for all s_k, p(s_k)^2 <= p(s_k)). Therefore, we can establish that our bound on the error rate lies in the range 1/2|X| <= Err* <= 1/2. Intuitively, in the case of a one-hot distribution, only one summary is ever possible, and therefore anything that belongs to the product of marginals also belongs to the joint---making the classifier's choices a random guess.
> > >
> > > We have checked again, and can confirm that our procedure gives us an upper bound on the error rate. As the error rate is proportional to the sum of eq (2) (the product of marginals over pairs X and R(X)), and the conditional is bounded from above by one, the sum of the marginal squared is an upper bound (including the conditional in the sum will only decrease the value of the sum).
> > >
> > > Lastly, it is unclear what is meant by a “sufficient statistic” in this case: could you provide a concrete example? Given the assumption of the problem (X being i.i.d. and a deterministic encoder), the max-info encoder must be invertible. A simple counter example (two i.i.d. samples mapping to the same feature vector) is trivially suboptimal.
> > >
> > > We have made modifications to the paper in the Conclusions section (as proposed) and added remarks about the bounds on Err*.
> > >
> > > Thank you once again for your comments!

---

> > > > ### Comment · AnonReviewer2 · 2018-12-03
> > > > **Regarding sufficient statistics**
> > > >
> > > > Thank you very much for the revision.
> > > >
> > > > Based on your formulations, the sufficient statistics is any transformation of $X^{(k)}$ which is sufficient to evaluate the likelihood $p(X^{(k)})$. For example, any injective mapping $\mathcal{R}$ results in a sufficient statistic. However, this condition is not necessary. The information monotonicity bound is tight under transformations corresponding to sufficient statistics: $KL(p(x|theta_1),p(x|theta_2)) \ge KL(p(f(x)|theta_1),p(f(x)|theta_2))$, where "=" holds iff $f(x)$ is a sufficient statistic meaning that under the transformation $f$ there is no loss of information.

---

### Official Review · AnonReviewer1 · 2018-10-31
**Idea is interesting; realization is graph-specific**

**Rating:** 5
**Confidence:** 4

**Review:**

This paper proposes an unsupervised approach to learning node representations. The basic steps are: (1) use an encoder E to learn node vectors, (2) use a readout function R to summarize node vectors into the graph vector, (3) use a scoring function D to score how much the node vectors are aligned with the graph vector, and (4) maximize the scores for the given graph meanwhile minimize the those from the negative distribution.

I feel that the idea is interesting; however, the paper is less well written and the realization of the idea has drawbacks as well.

1. Presentation of Section 3.2 can be improved. The proposed approach becomes clear only toward the end.

2. Naming and wording is misleading. The title and the whole paper use the wording "mutual information", whereas in reality, the loss function is a cross entropy.

3. In equation (1), it is unclear why the authors take expectation with respect to the distribution of graphs before summing the scores for one particular graph. Should the order of the expectation and summation be swapped?

4. The proposal is more like a framework than a specific method. The encoder and the negative distribution need to be separately designed for different graphs.

Good things about the proposal:

5. The downstream classification results are quite comparable to those of supervised methods (except for the PPI data).

6. The learned node representations possess a clear clustering structure (Figure 3).

Minor comments:

7. In the third paragraph of section 4.3, the authors state that "... for the GCN model in the fully supervised setting". GCN should be a semi-supervised method rather than a fully-supervised one.

8. In the last paragraph of section 4.3, what is a "randomly initialized graph convolutional network" and how is it different from the proposal?

---

> ### Author Response · Authors · 2018-11-20
> **Reply to AnonReviewer1**
>
> Firstly, thanks so much for your thorough review!
>
> Towards your comment about Section 3.2 and Equation 1, please see our global comment on mutual information and the JSD. We reorganised Sections 3.2 / 3.3 to include more concrete theoretical motivation from mutual information maximisation, and separate this from the specifics that are important for implementation.
>
> We agree with your comment that the GCN is in fact used in a semi-supervised setting (as not all nodes are labelled). What we referred to is that the learning objective is fully supervised (solely cross-entropy on the training nodes’ labels), and have revised the paper accordingly.
>
> We acknowledge your comment about our method having the traits of a framework, but claim that such features are a consequence of the current state of the art in graph neural networks, rather than any limitations of our methodology. Namely:
>
> - Similar to other recently proposed GCN methods, such as the DiffPool algorithm (Ying et al., NIPS 2018), we indeed are agnostic to the choice of the GCN layer. This is because graph convolutional networks are a very active area of research and we don't currently have a “catch-all” layer for all possible scenarios (e.g. transductive vs. inductive).
>
> - The fact that different high-level architectures are used is normal, and constitutes hyperparameter optimisation and/or relating the work to previous successful architectures.
>
> - Our negative distribution choice is, in fact, mostly uniform. We’d like to use different input graphs as negative examples (as DIM does), but this is only possible (with a limited pool of examples) for PPI. In all other case we use node-wise shuffling, which was demonstrably robust---and we also motivate this robustness with further studies in Appendix C.
>
>
> A randomly initialised graph convolutional network is basically the setting in which we set the number of training epochs to zero---i.e. we start with weights initialised according to Xavier initialisation, and then immediately proceed to use this as our encoder rather than performing any unsupervised training of the encoder.
>
> We thank you once again for your review, which has definitely helped make our paper’s contributions stronger!

---

> > ### Comment · AnonReviewer1 · 2018-11-23
> > **Comments after revision**
> >
> > I appreciate the authors' thorough response to the concerns (of all reviewers).  Reading the discussions as well as the revised paper, I would like to offer the following comments.
> >
> > - The theoretical justification on mutual information adds value to the paper, although strictly speaking, the loss function is a GAN-style estimation of the mutual information, rather than the Jensen-Shannon estimation. See Table 2 and eqn (8) of https://arxiv.org/abs/1606.00709
> >
> > - My main problem remains that the authors separately design encoders (and to a lesser extent, the negative distributions) for different graphs. It is true that state-of-the-art architectures differ as the data sets vary, for the counterpart (semi)supervised setting. But the authors are not using the counterpart architectures, either. For example, for Cora, Citeseer, and Pubmed, the authors use a one-layer GCN. Then, questions naturally arise: Is such an encoder optimal? What if one replaces the one-layer GCN by the usual GCN, or the Planetoid architecture (note that these architectures are used as supervised baselines in the experiments)? Similar questions also arise for the data sets Reddit and PPI. My point is, what is missing in this work is a guidance regarding how should one choose the encoder (and the negative distribution) given a graph. The guidance is important for practitioners who want to apply the framework.

---

> > > ### Author Response · Authors · 2018-11-30
> > > **Response to Comments**
> > >
> > > Thank you for following the discussion, and giving us further comments. To address them:
> > >
> > > For the optimal discriminator, the GAN and JSD objectives are directly proportional to each other, so in this limit the two discriminators actually optimize the exact same thing (this was pointed out in original GAN paper by Goodfellow et al.). However, the proportionality actually extends to the non-optimal discriminator: plugging in the activation function and convex conjugate into equation (7) in the f-GAN paper for JSD and GAN, makes these two versions of the Fenchel dual proportional as well. We have updated the paper to clarify this.
> > >
> > > While we do not perform a more exhaustive study of architectures, and our modifications are fairly small compared to common architectures used in other works, we can offer some further insights here based on our experiences.
> > >
> > > Overall, we found that using the “wide” architectures with a standard supervised loss quickly and drastically overfits, whereas increasing the depth for a DGI encoder had a less-pronounced effect. For the transductive datasets especially, increased depth could induce a 2-3% drop in accuracy, and the reason for this is likely because deeper GCNs have larger “receptive fields”. This is analogous to the number of random walk steps in other unsupervised approaches, which tend to not benefit from more than two steps. While we cannot say that these trends will hold in general, we now offer some concrete suggestions in the paper: i.e., that with the DGI loss function we often found benefits from employing wider, rather than deeper models.

---

### Official Review · AnonReviewer3 · 2018-11-02
**Solid work, will have high impact**

**Rating:** 9
**Confidence:** 4

**Review:**

This paper describes an approach for unsupervised learning of node features on a graph (with known structure), so that learned local representations represent community information that has high mutual info with a graph-level summary. The general idea is they apply InfoMax to graphs via graph convolutional networks (GCN), and report impressive results, including rivaling supervised learning methods for node classification. The 3 experiments are on paper topic classification, social network modeling, and protein classification.

The idea of using InfoMax with GCNs for unsupervised node learning is clever and timely, the technical contribution is solid, the experiments are executed well, and the paper is clear and easy to read.

---

> ### Author Response · Authors · 2018-11-20
> **Reply to AnonReviewer3**
>
> Thank you very much for the extremely kind review, and we are very glad you enjoyed the paper! We have made further updates to the paper---some details of which are outlined in our global comment above.

---

### Public Comment · ~Ming_Ding1 · 2018-09-28
**The experiments about transductive learning are incomplete**

I think the idea is very interesting. But in the experiments under transductive semi-supervised learning setting, some methods with better performance are missed, for example GAT(Velickovic et al., 2018) and even our GraphSGAN(Ding et al., 2018). I know that this method is actually inductive but you should at least cite and list above results.

---

> ### Author Response · Authors · 2018-09-29
> **Provided results**
>
> Hi Ming,
>
> Thanks for your comment and kind feedback!
>
> DGI performs unsupervised learning, so comparisons with supervised methods are inappropriate---the supervised methods we did include were the ones that used similar propagation rules as our encoders. In the transductive case, this was the GCN---as it uses an identical propagation rule (with one extra layer).
>
> We'll cite GraphSGAN as an indicator of the current supervised state-of-the-art in an updated version of the paper -- thanks!

---

### Public Comment · (anonymous) · 2018-11-09
**Needs more explanation on how this is different from Embedding Propagation work**

The work has similarity to Embedding Propagation (Duran et al, NIPS 2017) work, where the node's embedding is made similar to averaged 1-hop neighbors. DGI seems to have extended this idea into a framework. Where the neighbor's summary obtained with an averaging function is abstracted and called as readout function. The embeddings/feature (projection) has been now obtained with a GCN encoder.

Though, there is a mention of this work. I feel enough justice to the earlier work has not been given.

---

> ### Author Response · Authors · 2018-11-20
> **On the link to Embedding Propagation (EP)**
>
> Thank you for your comment pointing out similarities between our work and Embedding Propagation (EP). However, we believe that some of the similarities you mention might have been misattributed. To highlight:
>
> In DGI, we not only add graph convolutions, we also employ a very different objective function, where the goal is to maximize the mutual information between single node embeddings and a summary embedding of the entire graph. Thus, there are two key points of clarification regarding the fundamental difference between our work and EP:
>
> 1) the “readout” function in our work does not simply abstract away the neighborhood aggregation function in EP, since in our work the “readout” function embeds an entire graph;
>
> 2) the objective in our work maximizes the mutual information between a single node embedding and the summary graph embedding, whereas in EP they use a “one-step” random walk objective, which maximizes the similarity between a node embedding and the embedding of its local neighborhood.

---

### Public Comment · (anonymous) · 2018-11-09
**Authors should quote more recent results for PPI dataset**

It has already been noted that GraphSAGE models can achieve superior results (20-30% improvement) with a different hyperparameter setting on PPI than what was reported in the original paper. It is even mentioned in the Arxiv version of GraphSAGE. It is unfair to report such low scores and I also encourage the authors to update DGI's hyperparam setting correspondingly and report newer numbers.

---

> ### Author Response · Authors · 2018-11-09
> **More recent results for PPI are reported**
>
> Hello,
>
> Thank you for your comment!
>
> All reported improvements in PPI results concern solely the fully supervised setup; not the unsupervised one. And, indeed, this is the supervised result we report---namely, the avg. pooling architecture from the GaAN paper (Zhang et al., UAI 2018), which we report, is one example of a supervised result that substantially (30+%) improves on the supervised result reported in the original GraphSAGE paper (of 0.612).

---

### Author Response · Authors · 2018-11-20
**[Revision] On mutual information estimation and maximization and the cross-entropy**

We agree that the discussion on the connections between mutual information and the Jensen-Shannon Divergence / binary cross entropy was insufficient in our paper, and we have added further details in our revision.

We have added this intuition as motivation to our loss function in Section 3.3.

For a brief overview:

Let p(X, f(X)) be the joint distribution of the random variable X and f(X), a random variable corresponding to the transformation of X by a deterministic function, f. Also, let p(X) be an empirical distribution specified by a finite set of given samples, and let p(f(X)) be the marginal.

In our setting, we both train a classifier to distinguish between samples from p(X, f(X)) and from p(X) p(f(X))) and find the function f in F that minimizes the same classifier’s loss. In other words, we are looking for the functions f* in F that satisfy
f* = argmin_f argmin_c error(p(X, f(X)), p(X) p(f(X)); c),
and we use the binary cross-entropy (BCE) as a proxy for the classification error.

It is enough to show that optimal solutions to the above only contain functions f that are invertible, as the mutual information MI(X; f(X)) is invariant over invertible functions of F, and maximized for them. Because f is deterministic, and considering a discrete X, card(X) >= card(f(X)), where with non-invertible functions, this ordering is strict ( i.e., card(X) > card(f(X))). Given a mixture between the joint and the product of marginals, it can be shown that the optimal f (under the classification error) has card(X) = card(f(X)). This and the fact that f is deterministic implies there exists an inverse. Hence the f* that minimizes the classification error between the joint and the product of marginals in this setting also maximizes the mutual information MI(X;f(X)) = MI(X; X) = H(X).

---

### Meta-Review · Area_Chair1 · 2018-12-14
**borderline paper due to concerns remain about the thoroughness of the experiments**

**Confidence:** 3
**Recommendation:** Accept (Poster)

**Metareview:**

Because of strong support from two of the reviewers I am recommending accepting this paper. However, I believe reviewer 1's concerns should be taken seriously. Although I disagree with the reviewer that a general "framework" method is a bad thing, I agree with them that additional experiments would be valuable.